**Data Availability Statement:** The ethics committee of the Japanese Association of Neuro-Psychiatric Clinics set restrictions on data sharing because the data contain potentially identifying or sensitive

# Real-world clinical predictors of manic/ hypomanic episodes among outpatients with bipolar disorder

Keita Tokumitsu[1], Yasui-Furukori Norio[1,2]*, Naoto Adachi[3], Yukihisa Kubota[3], Yoichiro Watanabe[3], Kazuhira Miki[3], Takaharu Azekawa[3], Koji Edagawa[3], Eiichi Katsumoto[3], Seiji Hongo[3], Eiichiro Goto[3], Hitoshi Ueda[3], Masaki Kato[2,4], Atsuo Nakagawa[2,5], Toshiaki Kikuchi[2,5], Takashi Tsuboi[2,6], Koichiro Watanabe[2,6], Kazutaka Shimoda[1], Reiji Yoshimura[2,7]

1 Department of Psychiatry, Dokkyo Medical University School of Medicine, Tochigi, Japan, 2 The Japanese Society of Clinical Neuropsychopharmacology, Tokyo, Japan, 3 The Japanese Association of Neuro-Psychiatric Clinics, Tokyo, Japan, 4 Department of Neuropsychiatry, Kansai Medical University, Osaka, Japan, 5 Department of Neuropsychiatry, Keio University School of Medicine, Tokyo, Japan, 6 Department of Neuropsychiatry, Kyorin University School of Medicine, Tokyo, Japan, 7 Department of Psychiatry, University of Occupational and Environmental Health, Fukuoka, Japan

* furukori@dokkyomed.ac.jp

## Abstract

### Background

Bipolar disorder is a mental illness in which manic and depressive states are repeated, causing psychosocial dysfunction. Manic/hypomanic episodes cause problems with interpersonal, social and financial activities, but there is limited evidence regarding the predictors of manic/hypomanic episodes in real-world clinical practice.

### Methods

The multicenter treatment survey on bipolar disorder (MUSUBI) in Japanese psychiatric clinics was administered in an observational study that was conducted to accumulate evidence regarding bipolar disorder in real-world clinical practice. Psychiatrists were asked to complete a questionnaire about patients with bipolar disorder who visited 176 member clinics of the Japanese Association of Neuro-Psychiatric Clinics by conducting a retrospective medical record survey. Our study extracted baseline patient characteristics from September to October 2016, including comorbidities, mental status, duration of treatment, Global Assessment of Functioning (GAF) score, and pharmacological treatment details. We investigated the presence or absence of manic/hypomanic episodes over the course of one year from baseline to September-October 2017.

### Results

In total, 2231 participants were included in our study, 29.1% of whom had manic/hypomanic episodes over the course of one year from baseline. Binomial logistic regression analysis revealed that the presence of manic/hypomanic episodes was correlated with lower

patient information. Please contact the institutional review board of the ethics committee when requesting data. Contact information for our ethics committee: The institutional review board of the ethics committee of the Japanese Association of Neuro-Psychiatric Clinics; Shibuya-ku, Yoyogi 1-38-2, Tokyo Metropolis, Japan, Postal Code 151–0053, Phone +81-3-3320-1423.

**Funding:** This study was supported by a Ken Tanaka memorial research grant (grant numbers: 2016–2, 2017–4 and 2019–3). The funder had no role in the study design, the data collection and analysis, the decision to publish, or the preparation of the manuscript.

**Competing interests:** Yasui-Furukori has received grant/research support or honoraria from and has been a speaker for Dainippon-Sumitomo Pharma, Mochida Pharmaceutical, MSD, and Otsuka Pharmaceutical. Shimoda has received research support from Novartis Pharma, Dainippon Sumitomo Pharma, Astellas Pharma, Meiji Seika Pharma, Eisai, Pfizer, Otsuka Pharmaceutical, Daiichi Sankyo, and Takeda Pharmaceutical and honoraria from Eisai, Mitsubishi Tanabe Pharma, Takeda Pharmaceutical, Meiji Seika Pharma, Janssen Pharmaceutical, Shionogi, Dainippon Sumitomo Pharma, Daiichi Sankyo, and Pfizer. Yoshimura has received speaker honoraria from Eli Lilly, Janssen, Dainippon Sumitomo, Otsuka, Meiji, Pfizer and Shionogi. Kato has received grant funding from the Japan Society for the Promotion of Science, SENSHIN Medical Research Foundation and the Japan Research Foundation for Clinical Pharmacology and has received speaker honoraria from Dainippon-Sumitomo Pharma, Otsuka, Meiji-Seika Pharma, Eli Lilly, MSD K.K., GlaxoSmithKline, Pfizer, Janssen Pharmaceutical, Shionogi, Mitsubishi Tanabe Pharma, Takeda Pharmaceutical and Ono Pharmaceutical. Azegawa has received speaker honoraria from Eli Lilly, Otsuka Pharmaceutical, and Pfizer. Ueda has received manuscript fees or speaker honoraria from Eli Lilly, Janssen Pharmaceutical, Meiji Seika Pharma, Mitsubishi Tanabe Pharma, MSD, Otsuka Pharmaceutical, Pfizer, Sumitomo Dainippon Pharma, Takeda Pharmaceutical, and Yoshitomi Yakuhin. Edagawa has received speaker honoraria from Eli Lilly, Meiji Seika Pharma, Mitsubishi Tanabe Pharma, MSD, Otsuka Pharmaceutical, Pfizer, Sumitomo Dainippon Pharma, Kyowa and Yoshitomi Yakuhin. Katsumoto has received speaker honoraria from Daiichi Sankyo, Eisai, Eli Lilly, Janssen Pharmaceutical, Kyowa Pharmaceutical, Meiji Seika Pharma, Mitsubishi Tanabe Pharma, MSD, Otsuka Pharmaceutical, Pfizer, Sumitomo Dainippon Pharma, and UCB.

baseline GAF scores, rapid cycling, personality disorder, bipolar I disorder, and a mood state with manic or mixed features. Substance abuse was also a risk factor for manic episodes. There was no significant association between a baseline antidepressant prescription and manic/hypomanic episodes.

## Conclusions

In Japan, 29.1% of outpatients with bipolar disorder had manic/hypomanic episodes over the course of one year. Our study suggested that a low GAF score, rapid cycling, personality disorder, bipolar I disorder, substance abuse, and baseline mood state could be predictors of manic/hypomanic episodes. Based on our findings, an antidepressant prescription is not a predictor of manic/hypomanic episodes.

## Introduction

Bipolar disorder is an affective disorder that results in chronic repeated severe mood changes with manic and depressive episodes; it is associated with characteristic cognitive, physical, and behavioral impairments [1–3]. Worldwide, the estimated lifetime prevalence of bipolar disorder among adults is 2.4% [4]. The prevalence is similar between men and women [5]. The average onset age for bipolar I disorder is 18 years, while that for bipolar II disorder is 20 years [4]. Inappropriate behavior caused by mood changes often has negative impacts on social, financial, and occupational outcomes [2]. Patients with bipolar disorder are 20 times more likely to commit suicide than the general population and bipolar disorder is associated with a high suicide rate [6].

Depression in patients with bipolar disorder is associated with long-term morbidity and impaired function [7]. The manic state is associated with increased medical costs and inappropriate behavior that leads to social problems such as wasting money and having aggressive attitudes [6]. Previous studies have shown that mood stabilizers reduce the risk of manic-depressive episodes and that antidepressant monotherapy may increase the risk of manic episodes. The incidence of treatment-emergent mania is reported to be 11.8–30.9% depending on the study settings and the incidence of treatment-emergent mania was low with the combined prescription of lithium [8] and second-generation antipsychotics [9]. However, there is contradictory evidence about the effectiveness of antidepressants for bipolar disorder, and no consensus has yet been reached. The predictors of manic episodes in the real world remain unclear.

More than 90% of mood disorder patients in Japan are outpatients, and approximately 50% of them are treated at clinics belonging to the Japanese Association of Neuro-Psychiatric Clinics (JAPC) [10]. Given this context, a collaborative study (the MUlticenter treatment SUrvey on BIpolar disorder in Japanese psychiatric clinics, or MUSUBI) was performed by the JAPC and the Japanese Society of Clinical Neuropsychopharmacology (JSCNP) to accumulate evidence on the real-world treatment of bipolar disorder in clinical practice in Japan [10–14]. In the MUSUBI study, we found that antidepressants are prescribed to a high proportion of patients with bipolar disorder (40.9%) [10], but the risk of manic/hypomanic episodes associated with the use of antidepressants could not be fully elucidated because of the cross-sectional design of the study. Antidepressant-induced mania/hypomania is a controversial issue that requires further research in not only clinical studies in specific settings but also real-world

Kubota has received consultant fees from Pfizer and Meiji-Seika Pharma and speaker honoraria from Meiji-Seika Pharma, Eli Lilly, Janssen Pharmaceutical, Dainippon Sumitomo Pharma, Mitsubishi Tanabe Pharma, Yoshitomi Yakuhin, Otsuka Pharmaceutical, and Eisai. Goto has received manuscript fees or speaker honoraria from Eli Lilly, Janssen Pharmaceutical, Meiji Seika Pharma, Mitsubishi Tanabe Pharma, MSD, Otsuka Pharmaceutical, and Sumitomo Dainippon Pharma. Hongo has received manuscript fees or speaker honoraria from Eli Lilly, Janssen Pharmaceutical, Kyowa Pharmaceutical, Meiji Seika Pharma, Mitsubishi Tanabe Pharma, Otsuka Pharmaceutical, Pfizer, Shionogi, Sumitomo Dainippon Pharma, and Yoshitomi Yakuhin. Tsuboi has received consultant fees from Pfizer and speaker honoraria from Eli Lilly, Meiji-Seika Pharma, MSD, Janssen Pharmaceutical, Dainippon Sumitomo Pharma, Mitsubishi Tanabe Pharma, Yoshitomi Yakuhin, Mochida Pharmaceutical, Otsuka Pharmaceutical, Kyowa Pharmaceutical, and Takeda Pharmaceutical. Nakagawa has received lecture fees from Pfizer, Eli Lilly, Otsuka, Janssen Pharmaceutical, Mitsubishi Tanabe, Mochida, Dainippon Sumitomo and NTT Docomo and has served on advisory boards for Takeda, Meiji Seika and Tsumura. Kikuchi has received consultant fees from Takeda Pharmaceutical and the Center for Cognitive Behavioral Therapy and Training. Watanabe has received manuscript fees or speaker honoraria from Daiichi Sankyo, Eisai, Eli Lilly, GlaxoSmithKline, Janssen Pharmaceutical, Kyowa Pharmaceutical, Meiji Seika Pharma, Mitsubishi Tanabe Pharma, MSD, Otsuka Pharmaceutical, Pfizer, Shionogi, Sumitomo Dainippon Pharma, Takeda Pharmaceutical, and Yoshitomi Yakuhin; has received research/grant support from Astellas Pharma, Daiichi Sankyo, Eisai, MSD, Mitsubishi Tanabe Pharma, Meiji Seika Pharma, Otsuka Pharmaceutical, Pfizer, Shionogi, and Sumitomo Dainippon Pharma; and is a consultant for Eisai, Eli Lilly, Kyowa Pharmaceutical, Otsuka Pharmaceutical, Pfizer, Sumitomo Dainippon Pharma, Taisho Toyama Pharmaceutical, and Takeda Pharmaceutical.

clinical practice. Therefore, we investigated the occurrence of manic/hypomanic episodes during the one year after baseline, focusing not only on baseline clinical features but also on each class of antidepressant prescribed to patients with bipolar disorder.

## Subjects and methods

### Study design and subjects

The MUSUBI study was a retrospective study in which a questionnaire was administered at 176 outpatient clinics belonging to the JAPC, with baseline patient characteristics collected from September to October 2016 [6]. We investigated the occurrence of manic/hypomanic episodes over the course of one year from baseline to September-October 2017. Patients diagnosed with bipolar disorder based on the ICD-10 criteria [3] and treated at these clinics were included in this study.

### Study procedures

Clinical psychiatrists were asked to complete a semi-structured questionnaire on patients with bipolar disorder by performing a retrospective medical record survey. The questionnaire included patient characteristics (age, sex, height, weight, academic background, and occupational status), comorbidities, mental status, Global Assessment of Functioning (GAF) score, and details of pharmacological treatment as the baseline data. In addition, we assessed the occurrence of manic/hypomanic episodes over the course of the year after baseline.

### Statistical analysis

All statistical analyses were performed with EZR (Saitama Medical Center, Jichi Medical University, Saitama, Japan) [15], which is a graphical user interface for R (The R Foundation for Statistical Computing, Vienna, Austria, version 3.5.2). More precisely, it is a modified version of R Commander (version 2.5–1) that incorporates statistical functions that are frequently used in biostatistics.

All statistical tests were two-sided, with a significance level of 0.05. Demographic and clinical characteristics were analyzed using the chi-square test and the Mann-Whitney U test to identify differences between patients with and without manic/hypomanic episodes over the course of the year after baseline. Univariate logistic regression analyses were performed to assess the demographic and clinical features. All factors associated with the occurrence of manic/hypomanic episodes among the bipolar disorder patients were identified using binomial logistic regression with forced entry to avoid overlooking any potential associations. These factors included sex, body mass index (BMI), age at study entry, age at the onset of bipolar symptoms, current employment status, education level, mood stabilizer prescription (valproic acid, lithium carbonate, carbamazepine and lamotrigine), antipsychotic prescription, anxiolytic (benzodiazepines only) prescription, hypnotic prescription, intelligence quotient (IQ), GAF score, psychiatric comorbidities, personality disorder, developmental disorder, physical comorbidities, rapid cycling status, substance abuse, suicidal ideation, types of bipolar disorder and mood status.

The following variables were included: male = 1, female = 2; employed = 1, unemployed = 0; Special support education school = 0, Junior high school = 1, High school, vocational school = 2, Junior college, technical college = 3, University = 4, Master's degree or higher = 5; taking mood stabilizers = 1, not taking mood stabilizers = 0; taking antipsychotics = 1, not taking antipsychotics = 0; taking anxiolytics = 1, not taking anxiolytics = 0; taking hypnotics = 1, not taking hypnotics = 0; IQ (>85) = 0, IQ (85–71) = 1, IQ (<71) = 2; GAF (81–100) = 0, GAF

(61–80) = 1, GAF (41–60) = 2, GAF (<41) = 3 (the actual GAF scores and the dummy variable of GAF scores in this study are inversely correlated); psychiatric comorbidity = 1, no psychiatric comorbidity = 0; personality disorder = 1, no personality disorder = 0; developmental disorder = 1, no developmental disorder = 0; physical comorbidity = 1, no physical comorbidity = 0; rapid cycling = 1, no rapid cycling = 0; substance abuse = 1, no substance abuse = 0; bipolar I disorder = 1, bipolar II disorder = 2; and suicidal ideation = 1, no suicidal ideation = 0. Four mood states (depressive, mixed, remission and manic) were analyzed as nominal measures. A bipolar disorder not otherwise specified was listwise deleted; patients with serious physical conditions such as terminal cancer or intractable diseases were excluded. Cases with missing values in the questionnaire responses were excluded from the study. Other factors do not exclude patients in this study. In addition, we also examined the relationship between baseline antidepressant prescription and the occurrence of manic/hypomanic episodes during the one year after baseline.

### Ethics

This study was conducted in accordance with the Declaration of Helsinki and the Japanese Ethical Guidelines for Medical and Health Research Involving Human Subjects. Prior to the initiation of the study, the study protocol was reviewed and approved by the institutional review board of the ethics committee of JAPC and Dokkyo Medical University School of Medicine. Since this was a retrospective medical record survey, it was exempted from the requirement for informed consent; however, we released information on this research so that patients were free to opt out. Administrative permissions and licenses were acquired by our team to access the data used in our research. The ethics committee of the Japanese Association of Neuro-Psychiatric Clinics set restrictions on data sharing because the data contain potentially identifying or sensitive patient information. Please contact the institutional review board of the ethics committee when requesting data. Contact information for our ethics committee: The institutional review board of the ethics committee of the Japanese Association of Neuro-Psychiatric Clinics; Shibuya-ku, Yoyogi 1-38-2, Tokyo Metropolis, Japan, Postal Code 151–0053, Phone +81-3-3320-1423.

### Results

As baseline data, completed questionnaires on 3213 outpatients with bipolar disorder were returned from the 176 originally solicited outpatient facilities. A total of 982 outpatients were excluded with listwise deletions. Thus, data on a total of 2231 patients were included in this study. The characteristics of the subjects are shown in Table 1.

The proportion of patients who experienced manic/hypomanic episodes over the course of the year after baseline was 29.1% (694/2231). The results of the univariate analysis are shown in Table 1. Because 23 comparisons were made, a Bonferroni correction was applied, yielding a corrected significance criterion of p <0.0022. Based on this threshold, the occurrence of manic/hypomanic episodes significantly differed among the groups stratified by GAF scores, rapid cycler status, suicidal ideation, antipsychotic prescription, employment status, personality disorder, psychotic symptoms, psychiatric comorbidity, types of bipolar disorders, developmental disorder, mood status, substance abuse and hypnotic prescription.

A binomial logistic regression analysis revealed that the group of patients with manic/hypomanic episodes during the year after baseline had significantly lower GAF scores (odds ratio; OR [95% CI] = 1.54 [1.30–1.83], p<0.001), a higher proportion of patients with rapid cycling (OR = 3.86 [2.84–5.24], p<0.001), a higher proportion of patients with personality disorders (OR = 1.80 [1.02–3.17], p = 0.042), a higher proportion of patients with bipolar I disorder

**Table 1. Demographic and clinical data for participants.**

| Factors | Dummy variable | Total (N = 2231) | manic/hypomanic episode for the next one year | | p-value | Scale | Statistical method |
|---|---|---|---|---|---|---|---|
| | | | manic/hypomanic episodes (-) (N = 1537) | manic/hypomanic episodes(+) (N = 694) | | | |
| Gender | | | | | 0.02 | nominal scale | chi-square test |
| Male: n(%) | 1 | 1025 (45.9) | 732 (47.6) | 293 (42.2) | | | |
| Female: n(%) | 2 | 1206 (54.1) | 805 (52.4) | 401 (57.8) | | | |
| BMI (median [range]) | | 22.90 [14.50, 45.40] | 22.90 [14.50, 45.40] | 22.90 [14.60, 39.60] | 0.068 | interval scale | Mann-Whitney U-test |
| Age at study entry: (median [range]) | | 50.00 [13.00, 94.00] | 50.00 [15.00, 90.00] | 51.00 [13.00, 94.00] | 0.584 | interval scale | Mann-Whitney U-test |
| Age at onset of bipolar symptoms: (median [range]) | | 34.00 [10.00, 81.00] | 34.00 [12.00, 81.00] | 33.00 [10.00, 78.00] | 0.279 | interval scale | Mann-Whitney U-test |
| GAF score: n(%) | | | | | <0.001 | ordinal scale | Mann-Whitney U-test |
| 81–100 | 0 | 757 (33.9) | 620 (40.3) | 137 (19.7) | | | |
| 61–80 | 1 | 1019 (45.7) | 686 (44.6) | 333 (48.0) | | | |
| 41–60 | 2 | 399 (17.9) | 206 (13.4) | 193 (27.8) | | | |
| 1–40 | 3 | 56 (2.5) | 25 (1.6) | 31 (4.5) | | | |
| Current work status: n(%) | 0 | 744 (33.3) | 464 (30.2) | 280 (40.3) | <0.001 | nominal scale | chi-square test |
| | 1 | 1487 (66.7) | 1073 (69.8) | 414 (59.7) | | | |
| Intelligence Quotient, IQ: n (%) | | | | | 0.391 | ordinal scale | Mann-Whitney U-test |
| > 85 | 0 | 2140 (95.9) | 1479 (96.2) | 661 (95.2) | | | |
| 85–71 | 1 | 73 (3.3) | 48 (3.1) | 25 (3.6) | | | |
| < 71 | 2 | 18 (0.8) | 10 (0.7) | 8 (1.2) | | | |
| Educational level: n(%) | | | | | 0.086 | ordinal scale | Mann-Whitney U-test |
| Special support education school | 0 | 7 (0.3) | 5 (0.3) | 2 (0.3) | | | |
| Junior high school | 1 | 102 (4.6) | 67 (4.4) | 35 (5.0) | | | |
| High school, vocational school | 2 | 1037 (46.5) | 709 (46.1) | 328 (47.3) | | | |
| Junior college, technical college | 3 | 202 (9.1) | 131 (8.5) | 71 (10.2) | | | |
| University | 4 | 806 (36.1) | 580 (37.7) | 226 (32.6) | | | |
| Master's degree or higher | 5 | 77 (3.5) | 45 (2.9) | 32 (4.6) | | | |
| Rapid cycler: n (%) | 0 | 1973 (88.4) | 1450 (94.3) | 523 (75.4) | <0.001 | nominal scale | chi-square test |
| | 1 | 258 (11.6) | 87 (5.7) | 171 (24.6) | | | |
| Suicidal ideation: n (%) | 0 | 1990 (89.2) | 1404 (91.3) | 586 (84.4) | <0.001 | | |
| | 1 | 241 (10.8) | 133 (8.7) | 108 (15.6) | | | |
| Mood stabilizer prescription: n (%) | 0 | 349 (15.6) | 253 (16.5) | 96 (13.8) | 0.129 | nominal scale | chi-square test |
| | 1 | 1882 (84.4) | 1284 (83.5) | 598 (86.2) | | | |
| Antidepressants prescription: n (%) | 0 | 1300 (58.3) | 872 (56.7) | 428 (61.7) | 0.032 | nominal scale | chi-square test |
| | 1 | 931 (41.7) | 665 (43.3) | 266 (38.3) | | | |
| Antipsychotics prescription: n (%) | 0 | 1070 (48.0) | 776 (50.5) | 294 (42.4) | <0.001 | nominal scale | chi-square test |
| | 1 | 1161 (52.0) | 761 (49.5) | 400 (57.6) | | | |
| Anxiolytics prescription: n (%) | 0 | 1418 (63.6) | 974 (63.4) | 444 (64.0) | 0.82 | nominal scale | chi-square test |

*(Continued)*

**Table 1.** (Continued)

| Factors | Dummy variable | Total (N = 2231) | manic/hypomanic episode for the next one year | | p-value | Scale | Statistical method |
|---|---|---|---|---|---|---|---|
| | | | manic/hypomanic episodes (-) (N = 1537) | manic/hypomanic episodes(+) (N = 694) | | | |
| | 1 | 813 (36.4) | 563 (36.6) | 250 (36.0) | | | |
| Hypnotics prescription: n (%) | 0 | 897 (40.2) | 655 (42.6) | 242 (34.9) | 0.001 | nominal scale | chi-square test |
| | 1 | 1334 (59.8) | 882 (57.4) | 452 (65.1) | | | |
| Physical comorbidity: n (%) | 0 | 1555 (69.7) | 1091 (71.0) | 464 (66.9) | 0.056 | nominal scale | chi-square test |
| | 1 | 676 (30.3) | 446 (29.0) | 230 (33.1) | | | |
| Personality disorder: n (%) | 0 | 2119 (95.0) | 1478 (96.2) | 641 (92.4) | <0.001 | nominal scale | chi-square test |
| | 1 | 112 (5.0) | 59 (3.8) | 53 (7.6) | | | |
| Psychotic symptoms: n (%) | 0 | 2080 (93.2) | 1455 (94.7) | 625 (90.1) | <0.001 | nominal scale | chi-square test |
| | 1 | 151 (6.8) | 82 (5.3) | 69 (9.9) | | | |
| Psychiatric comorbidity: n (%) | 0 | 1855 (83.1) | 1314 (85.5) | 541 (78.0) | <0.001 | nominal scale | chi-square test |
| | 1 | 376 (16.9) | 223 (14.5) | 153 (22.0) | | | |
| Types of bipolar disorders: n (%) | | | | | <0.001 | nominal scale | chi-square test |
| Bipolar I disorder | 1 | 794 (35.6) | 464 (30.2) | 330 (47.6) | | | |
| Bipolar II disorder | 2 | 1437 (64.4) | 1073 (69.8) | 364 (52.4) | | | |
| Developmental disorder: n (%) | 0 | 2110 (94.6) | 1472 (95.8) | 638 (91.9) | <0.001 | nominal scale | chi-square test |
| | 1 | 121 (5.4) | 65 (4.2) | 56 (8.1) | | | |
| Substance abuse: n (%) | 0 | 2117 (94.9) | 1482 (96.4) | 635 (91.5) | <0.001 | nominal scale | chi-square test |
| | 1 | 114 (5.1) | 55 (3.6) | 59 (8.5) | | | |
| Mood status: n(%) | | | | | <0.001 | nominal scale | chi-square test |
| Depressive state | | 885 (39.7) | 617 (40.1) | 268 (38.6) | | | |
| Manic state | | 189 (8.5) | 68 (4.4) | 121 (17.4) | | | |
| Remission | | 942 (42.2) | 759 (49.4) | 183 (26.4) | | | |
| Mixed state | | 215 (9.6) | 93 (6.1) | 122 (17.6) | | | |

Abbreviations, BMI: Body Mass Index, GAF: Global Assessment Functioning.

Because 23 comparisons were made, a Bonferroni correction was applied, yielding a corrected significance criterion of p <0.0022.

rather than bipolar II disorder (OR = 0.55 [0.44–0.68], p<0.001) and a higher rate of patients with substance abuse disorders than those who did not suffer manic episodes. The groups in a manic state (OR = 4.54 [3.13–6.58], p<0.001) and a mixed state (OR = 2.32 [1.02–2.44], p = 0.039) had significantly higher proportions of patients with manic/hypomanic episodes than the group in a depressive state (Table 2). In a binomial logistic regression analysis with forced entry, multicollinearity between variables was not observed.

We also analyzed the associations between baseline antidepressant prescriptions and the occurrence of manic/hypomanic episodes. Antidepressants were classified into selective serotonin reuptake inhibitors (SSRIs), serotonin–norepinephrine reuptake inhibitors (SNRIs), noradrenergic and specific serotonergic antidepressants (NaSSAs), serotonin antagonists and

**Table 2. Binomial logistic regression analysis of baseline factors for manic/hypomanic episodes during the year after baseline.**

| Factor | Odds ratio (95% CI) | p-value |
|---|---|---|
| Gender | 1.21 (0.97–1.51) | 0.096 |
| BMI | 1.01 (0.98–1.04) | 0.570 |
| Age at study entry | 1.00 (0.99–1.01) | 0.600 |
| Age at onset of bipolar symptoms | 1.00 (0.99–1.02) | 0.480 |
| Current work status | 0.91 (0.72–1.14) | 0.400 |
| Educational level | 1.08 (0.98–1.20) | 0.120 |
| Mood stabilizer prescription | 1.07 (0.81–1.43) | 0.630 |
| Antipsychotics prescription | 1.08 (0.87–1.33) | 0.490 |
| Anxiolytics prescription | 0.94 (0.76–1.17) | 0.590 |
| Hypnotics prescription | 1.03 (0.84–1.28) | 0.760 |
| Antidepressants prescription | 0.87 (0.70–1.08) | 0.190 |
| IQ | 0.97 (0.65–1.47) | 0.900 |
| GAF score | 1.54 (1.30–1.83) | <0.001 |
| Psychiatric comorbidity | 0.85 (0.56–1.30) | 0.450 |
| Personality disorder | 1.80 (1.02–3.17) | 0.042 |
| Developmental disorder | 1.52 (0.87–2.66) | 0.140 |
| Physical comorbidity | 1.04 (0.83–1.30) | 0.750 |
| Rapid cycler | 3.86 (2.84–5.24) | <0.001 |
| Substance abuse | 1.58 (1.02–2.44) | 0.039 |
| Suicidal ideation | 1.13 (0.81–1.57) | 0.480 |
| Psychotic symptoms | 1.02 (0.68–1.52) | 0.930 |
| Types of bipolar disorders | 0.55 (0.44–0.68) | <0.001 |
| Mood status (vs Depressive state) | | |
| Manic state | 4.54 (3.13–6.58) | <0.001 |
| Remission | 0.88 (0.66–1.15) | 0.340 |
| Mixed state | 2.32 (1.65–3.26) | <0.001 |

Abbreviations, BMI: Body Mass Index, IQ: Intelligence Quotient, GAF: Global Assessment Functioning, CI: confidence interval.

reuptake inhibitors (SARIs), and tricyclic antidepressants (TeCAs), and baseline mood status was stratified and analyzed. There was no significant association between a baseline antidepressant prescription and the occurrence of manic/hypomanic episodes during the one year after baseline (Table 3).

## Discussion

Our study is the first multicenter study in Japan to investigate the predictors of manic/hypomanic episodes in outpatients with bipolar disorder. Importantly, in the binomial logistic regression analysis, we found that low baseline GAF scores, rapid cycling status, personality disorders, bipolar I disorder, substance abuse, and baseline mood states were predictors of manic/hypomanic episodes over the course of the year after baseline. This result cannot be generalized because the analysis was limited to Japanese outpatients, but manic/hypomanic episodes in patients with bipolar disorder can be predicted by considering multiple factors. Similarly, a previous study named the Systematic Treatment Enhancement Program for Bipolar Disorder (STEP-BD) reported that baseline manic state, rapid cycling status, substance abuse and suicide attempts were predictors of manic episodes [16]. Therefore, it is thought

**Table 3. The associations between baseline antidepressant prescriptions and the occurrence of manic/hypomanic episodes.**

| Factor | manic/hypomanic episodes (-) | manic/hypomanic episodes (+) | p-value |
|---|---|---|---|
| | n = 1537 | n = 694 | |
| NaSSA (%) | 114 (7.4) | 40 (5.8) | 0.182 |
| SARI (%) | 81 (5.3) | 37 (5.3) | 1.000 |
| SNRI (%) | 180 (11.7) | 62 (8.9) | 0.060 |
| SSRI (%) | 277 (18.0) | 125 (18.0) | 1.000 |
| TCA (%) | 123 (8.0) | 51 (7.3) | 0.654 |
| TeCA (%) | 45 (2.9) | 17 (2.4) | 0.619 |

Abbreviations, NaSSA: Noradrenergic and Specific Serotonergic Antidepressant, SARI:
serotonin antagonist and reuptake inhibitors, SNRI: Serotonin Noradrenaline Reuptake
Inhibitor, SSRI: Selective Serotonin Reuptake Inhibitor, TCA: Tricyclic Antidepressants, TeCA: Tetracyclic
Antidepressants.

that manic episodes in patients with bipolar disorder can be predicted by combining certain factors regardless of culture or ethnicity.

In our study, the effect sizes of baseline manic state, mixed feature state and rapid cycling status were medium, indicating that they are relatively important predictors. Both the manic state and mixed feature state are characterized by manic features. It seems that a manic episode increases the risk of a subsequent manic episode. Interestingly, a review of placebo-controlled trials for bipolar disorder suggested that the baseline mood state of patients with bipolar disorder who entered the clinical trial was a predictor of their subsequent prognosis [17]. When the baseline state is a manic state, manic/hypomanic episodes are more likely to occur during the observation period [17]. Additionally, rapid cycling is specific to patients with bipolar disorder who have manic episodes more than four times per year. Past studies have shown that if rapid cycling is observed at baseline, there is an increased risk of subsequent manic episodes. [14, 18]. Rapid cycling is also associated with increased alcohol abuse and suicide, indicating that careful monitoring is needed [19].

Our study showed that patients with bipolar disorder with personality disorder were at a significantly elevated risk of experiencing mania/hypomanic episodes over the next year. The relationship between personality disorder and bipolar disorder has been discussed, especially in terms of the potential for comorbid borderline personality disorder and bipolar disorder [20]. It has been reported that approximately 1 in 5 patients with bipolar disorder have been diagnosed with comorbidities of borderline personality disorder [21]. Patients with bipolar disorder in real-world clinical settings commonly present with personality disorders [22]. A meta-analysis of 13 studies revealed that among 1101 outpatients and inpatients with bipolar disorder, at least one personality disorder was present in 42% [23]. In addition, compared to bipolar patients without a personality disorder, patients with comorbid personality disorders have a more severe course of illness, including a decreased likelihood of recovering from mood episodes [22, 24–25], worse psychosocial functioning and more suicide attempts [22]. Patients with borderline personality disorder often have difficulty controlling their emotions and impulses, which can lead to violent behavior or mood swings that are difficult to distinguish from manic episodes. In addition, due to unstable interpersonal relationships, anxiety, suicide attempts, and self-harm may occur, which may be perceived as a depressive episode. In this way, the phenomenological distinction between bipolar disorder and personality disorders may be difficult because of the overlap between the features of personality disorder and the diagnostic criteria for mood episodes [26]. A previous study reported that nearly 40% of

borderline personality disorder patients were found to have a misdiagnosis of bipolar disorder [27], whereas other studies reported an even higher rate of the overdiagnosis of bipolar disorder [28]; therefore, a careful assessment is needed. Our study suggests that personality disorder may make it difficult for patients with bipolar disorder to maintain remission. Because manic episodes of bipolar disorder and the symptoms of personality disorder may affect each other, more careful follow-up is needed to achieve treatment success.

Previous reports have been published on the association between substance use disorders and bipolar disorder [26, 29]. The risk of bipolar disorder onset has been reported to increase with substance abuse such as alcohol abuse [6, 30]. In particular, the consumption of alcohol by patients with bipolar disorder increases their risk of manic episodes [30]. Some studies have shown that substance abuse is more likely to be associated with bipolar disorder, while others warn that patients with substance abuse are more likely to be overdiagnosed with bipolar disorder [31]. Substance abuse is also known to cause manic or depressive symptoms resembling those of bipolar disorder [26]. However, there is insufficient real-world evidence from patients with bipolar disorder with substance abuse disorders. We have shown that patients with bipolar disorder with substance abuse disorders are at a significantly elevated risk of experiencing a manic episode within a year. This is essential evidence that suggests that the treatment of comorbidities is important for patients with bipolar disorder.

We also found that when the baseline GAF score was low, manic/hypomanic episodes were more likely to occur in the next year. The DSM-5 stipulates that "the mood disturbance must be sufficiently severe to cause marked impairment in social or occupational functioning" for the diagnosis of manic episodes [32]. It is important to note that our study revealed that poor baseline social adaptation was a predictor of manic episodes. A previous study revealed that patients who had suffered more manic episodes had more cognitive dysfunction in the areas of verbal learning and memory [33]. Importantly, repeated and prolonged manic states impair cognitive function [34], and cognitive impairment itself is a predictor of poor prognosis in patients with bipolar disorder [35]. Treatment for cognitive dysfunction may improve the quality of life for patients with bipolar disorder by strengthening their academic performance, work capacity, and social relationships [6].

Our study indicated that patients with bipolar II disorder had a lower risk of developing manic/hypomanic episodes within a year than those with bipolar I disorder (OR = 0.55). In a previous 13-year long-term observational study, patients with bipolar II disorder were symptomatically ill in 53.9% of the weeks of follow-up. Patients had symptoms of depression in 50.3% of the weeks of follow-up and hypomania in 1.3% of the weeks of follow-up [36]. Patients with bipolar I disorder were reported to be symptomatically ill in 47.3% of the weeks of follow-up, with depressive symptoms in 31.9% of the weeks of follow-up and manic or hypomanic symptoms in 8.9% of the weeks of follow-up [37]. These studies found that bipolar II is characterized by a longer symptom duration (50.3% vs 47.3%), but manic episodes occur more frequently in bipolar I (8.9% vs 1.3%). The content of these previous studies is consistent with our findings that patients with bipolar I at baseline were more likely to have a manic/hypomanic episode over the course of the next year.

Interestingly, no significant relationship was found between an antidepressant prescription at baseline and the occurrence of manic/hypomanic episodes over the course of the next year. The results were similar when stratified by the baseline mood state. In addition, although there was no significant difference, the frequency of manic/hypomanic episodes tended to be lower in the group of patients prescribed SNRIs. Our study also demonstrated that SNRIs were the most commonly prescribed antidepressants among patients with bipolar disorder. The results of this study suggest the tolerability of antidepressants, particularly SNRIs, in patients with bipolar disorder. Recently, a systematic review and network meta-analysis of the efficacy and

tolerability of pharmacological treatments for the treatment of acute bipolar depression was published; however, it did not include studies involving SNRIs [38]. The prescription of antidepressants for bipolar disorder remains a controversial issue. In particular, the risk of conversion to a manic state because of the use of antidepressants has been repeatedly reported [39, 40]. However, a recent real-world study reported that antidepressants are used by a large proportion of patients with bipolar disorder [39, 41–43]. While the treatment guidelines emphasize the high risks associated with tricyclic antidepressant monotherapy, they describe the effectiveness of antidepressant combination therapy for the treatment of acute depression and maintenance of remission [44]. In a study that analyzed the STEP-BD sample, antidepressant treatment was not found to be a primary risk factor for conversion from depression to mania in patients with bipolar disorder [16]. The presence of a manic episode before the onset of depression in patients with bipolar disorder did not significantly predict the response to antidepressants [45]. Recently, single-agent antidepressant prescriptions for bipolar II patients have also been accepted [46]. Bipolar disorder involves a long duration of depression, so the use of antidepressants may be acceptable if the risks and benefits of each individual case are carefully considered [46, 47]; however, there is insufficient high-quality evidence regarding the use of antidepressants by patients with bipolar disorder, and further supporting investigations are needed in the future.

## Limitations

Our study only covers outpatients, and information about inpatients remains unknown. Therefore, it does not reflect the entire population with bipolar disorder. In addition, since the study was conducted only in clinics that belonged to the Japanese Association of Neuro-Psychiatric Clinics and consented to participate in the study, there is a possibility that the composition of participants was biased. In addition, patient selection was not randomized and this was a retrospective study, which could have also led to selection bias. The results may differ in the long term, as follow-up in this study was limited to one year. Regarding the predictors of manic episodes, the presence or absence of antidepressants was analyzed based on the existence of a prescription at baseline. The durations of the prescriptions for various agents were not investigated. This is a limitation of our research, and further study is needed. Furthermore, we did not consider the severity of manic episodes and did not distinguish between manic and hypomanic episodes. We did not determine the exact number of people who experienced manic episodes (not hypomanic episodes). Therefore, the relationships between the severity of mania and the independent variables are unclear.

In addition, a previous review suggested problems with polypharmacy for patients with bipolar disorder [48]. Previously, our research group reported polypharmacy in a series of collaborative studies, and we found that the severity of illness and an intractable disease course were significantly associated with the number of psychotropic drugs in real-world clinical settings [13]. We performed multivariate analysis with the presence or absence of various psychotropic drugs (mood stabilizers, antipsychotics, antidepressants, hypnotics) as independent variables in the present study. In addition to these factors, adding the presence or absence of polypharmacy of the above drugs as an independent variable causes potential statistically multicollinear problems because each factor becomes a nested structure. For this reason, the risk of manic/hypomanic episodes due to polypharmacy was not fully understood.

Personality disorders include a variety of clinical conditions, but this study did not subdivide them, such as classifying clusters of personality disorders. Moreover, the odds ratio may not be adequately approximated to the risk ratio because the event rate of manic/hypomanic episodes was greater than 10% [49].

## Conclusions

In Japan, 29.1% of outpatients with bipolar disorder had manic/hypomanic episodes over the course of one year. Our study suggests that a low GAF score, rapid cycling, personality disorder, bipolar I disorder, substance abuse, and baseline mood state could be predictors of manic/hypomanic episodes. Based on our findings, antidepressant prescription is not a predictor of manic/hypomanic episodes.

## Acknowledgments

The authors thank the following psychiatrists belonging to the Japanese Association of Neuro-Psychiatric Clinics: Dr. Kazunori Otaka, Dr. Satoshi Terada, Dr. Tadashi Ito, Dr. Munehide Tani, Dr. Atsushi Satomura, Dr. Hiroshi Sato, Dr. Hideki Nakano, Dr. Yoichi Nakaniwa, Dr. Eiichi Hirayama, Dr. Keiichi Kobatake, Dr. Koji Tanaka, Dr. Mariko Watanabe, Dr. Shiguyuki Uehata, Dr. Asana Yuki, Dr. Nobuko Akagaki, Dr. Michie Sakano, Dr. Akira Matsukubo, Dr. Yukihisa Kibota, Dr. Yasuyuki Inada, Dr. Hiroshi Oyu, Dr. Tsuneo Tsubaki, Dr. Tatsuji Tamura, Dr. Shigeki Akiu, Dr. Atsuhiro Kikuchi, Dr. Keiji Sato, Dr. Toshihiko Lee, Dr. Kazuyuki Fujita, Dr. Fumio Handa, Dr. Hiroyuki Karasawa, Dr. Kazuhiro Nakano, Dr. Kazuhiro Omori, Dr. Seiji Tagawa, Dr. Daisuke Maruno, Dr. Hiroaki Furui, Dr. You Suzuki, Dr. Takeshi Fujita, Dr. Yukimitsu Hoshino, Dr. Kikuko Ota, Dr. Akira Itami, Dr. Kenichi Goto, Dr. Norio Okamoto, Dr. Yoshiaki Yamano, Dr. Kiichiro Koshimune, Dr. Junko Matsushita, Dr. Takatsugu Nakayama, Dr. Kazuyoshi Takamuki, Dr. Nobumichi Sakamoto, Dr. Miho Shimizu, Dr. Muneo Shimura, Dr. Norio Kawase, Dr. Ryouhei Takeda, Dr. Takuya Hirota, Dr. Hideko Fujii, Dr. Riichiro Narabayashi, Dr. Yutaka Fujiwara, Dr. Junkou Sato, Dr. Kazu Kobayashi, Dr. Yuko Urabe, Dr. Miyako Oguru, Dr. Osamu Miura, Dr. Yoshio Ikeda, Dr. Hidemi Sakamoto, Dr. Yosuke Yonezawa, Dr. Makoto Nakamura, Dr. Yoichi Takei, Dr. Toshimasa Sakane, Dr. Kiyoshi Oka, Dr. Kyoko Tsuda, Dr. Yasushi Furuta, Dr. Yoshio Miyauchi, Dr. Keizo Hara, Dr. Misako Sakamoto, Dr. Shigeki Masumoto, Dr. Yasuhiro Kaneda, Dr. Yoshiko Kanbe, Dr. Masayuki Iwai, Dr. Naohisa Waseda, Dr. Nobuhiko Ota, Dr. Takahiro Hiroe, Dr. Ippei Ishii, Dr. Hideki Koyama, Dr. Terunobu Otani, Dr. Osamu Takatsu, Dr. Takashi Ito, Dr. Norihiro Marui, Dr. Toru Takahashi, Dr. Tetsuro Oomori, Dr. Toshihiko Fukuchi, Dr. Kazumichi Egashira, Dr. Shigemitsu Hayashi, Dr. Kiyoshi Kaminishi, Dr. Ryuichi Iwata, Dr. Satoshi Kawaguchi, Dr. Kazuko Miyauchi, Dr. Yoshinori Morimoto, Dr. Kunihiko Kawamura, Dr. Hirohisa Endo, Dr. Yasuo Imai, Dr. Eri Kohno, Dr. Aki Yamamoto, Dr. Naomi Hasegawa, Dr. Sadamu Toki, Dr. Hideyo Yamada, Dr. Hiroyuki Taguchi, Dr. Hiroshi Yamaguchi, Dr. Hiroki Ishikawa, Dr. Sakura Abe, Dr. Kazuhiro Uenoyama, Dr. Kazunori Koike, Dr. Mikako Oyama, Dr. Yoshiko Kamekawa, Dr. Michihito Matsushima, Dr. Ken Ueki, Dr. Sintaro Watanabe, Dr. Tomohide Igata, Dr. Yoshiaki Higashitani, Dr. Eiichi Kitamura, Dr. Junko Sanada, Dr. Takanobu Sasaki, Dr. Kazuko Eto, Dr. Ichiro Nasu, Dr. Kenichiro Sinkawa, Dr. Yukio Oga, Dr. Michio Tabuchi, Dr. Daisuke Tsujimura, Dr. Tokunai Kataoka, Dr. Kyohei Noda, Dr. Nobuhiko Imato, Dr. Ikuko Nitta, Dr. Yoshihiro Maruta, Dr. Satoshi Seura, Dr. Toru Okumura, Dr. Osamu Kino, Dr. Tomoko Ito, Dr. Ryuichi Iwata, Dr. Wataru Konno, Dr. Toshio Nakahara, Dr. Masao Nakahara, Dr. Hiroshi Yamamura, Dr. Masatoshi Teraoka, Dr. Eiichiro Goto, Dr. Masato Nishio, Dr. Miwa Mochizuki, Dr. Tsuneo Saitoh, Dr. Tetsuharu Kikuchi, Dr. Chika Higa, Dr. Hiroshi Sasa, Dr. Yuichi Inoue, Dr. Muneyoshi Yamada, Dr. Yoko Fujioka, Dr. Kuniaki Maekubo, Dr. Hiroaki Jitsuiki, Dr. Toshihito Tsutsumi, Dr. Yasumasa Asanobu, Dr. Seiji Inomata, Dr. Kazuhiro Kodama, Dr. Aikihiro Takai, Dr. Asako Sanae, Dr. Shinichiro Sakurai, Dr. Kazuhide Tanaka, Dr. Masahiko Shido, Dr. Haruhisa Ono, Dr. Wataru Miura, Dr. Yukari Horie, Dr. Tetso Tashiro, Dr. Tomohide Mizuno, Dr. Naohiro Fujikawa, Dr. Hiroshi Terada, Dr. Kenji Taki, Dr. Kyoko Kyotani, Dr. Masataka Hatakoshi, Dr. Katsumi

Ikeshita, Dr. Keiji Kaneta, Dr. Ritsu Shikiba, Dr. Tsuyoshi Iijima, Dr. Masaru Yoshimura, Dr. Naoto Adachi, Dr. Masumi Ito, Dr. Shunsuke Murata, Dr. Mio Mori, and Dr. Toshio Yokouchi.

## Author Contributions

**Conceptualization:** Yasui-Furukori Norio, Naoto Adachi, Yukihisa Kubota, Takaharu Azekawa, Koji Edagawa, Eiichi Katsumoto, Seiji Hongo, Eiichiro Goto, Masaki Kato, Takashi Tsuboi, Koichiro Watanabe, Reiji Yoshimura.

**Data curation:** Hitoshi Ueda.

**Funding acquisition:** Yoichiro Watanabe, Kazuhira Miki, Eiichiro Goto.

**Investigation:** Keita Tokumitsu.

**Methodology:** Naoto Adachi, Yukihisa Kubota, Atsuo Nakagawa, Takashi Tsuboi.

**Supervision:** Yasui-Furukori Norio, Yoichiro Watanabe, Kazuhira Miki, Takaharu Azekawa, Koji Edagawa, Eiichi Katsumoto, Seiji Hongo, Eiichiro Goto, Masaki Kato, Toshiaki Kikuchi, Koichiro Watanabe, Kazutaka Shimoda, Reiji Yoshimura.

**Validation:** Atsuo Nakagawa, Toshiaki Kikuchi.

**Writing – original draft:** Keita Tokumitsu.

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
