## [Decision Letter · Decision Letter 0]

18 Aug 2021

PONE-D-21-18468

Real-world clinical predictors of manic episodes among outpatients with bipolar disorder.

PLOS ONE

Dear Dr. Norio,

Thank you for submitting your manuscript to PLOS ONE. After careful consideration, we feel that it has merit but does not fully meet PLOS ONE’s publication criteria as it currently stands. Therefore, we invite you to submit a revised version of the manuscript that addresses the points raised during the review process.

We look forward to receiving your revised manuscript.

Kind regards,

Michele Fornaro

Academic Editor

PLOS ONE

Journal Requirements:

- https://www.nature.com/articles/nrdp20188

- https://bmcpsychiatry.biomedcentral.com/articles/10.1186/s12888-020-02967-5

In your revision ensure you cite all your sources (including your own works), and quote or rephrase any duplicated text outside the methods section. Further consideration is dependent on these concerns being addressed.

Yasui-Furukori has received grant/research support or honoraria from and has been a speaker for Dainippon-Sumitomo Pharma, Mochida Pharmaceutical, MSD, and Otsuka Pharmaceutical. Shimoda has received research support from Novartis Pharma, Dainippon Sumitomo Pharma, Astellas Pharma, Meiji Seika Pharma, Eisai, Pfizer, Otsuka Pharmaceutical, Daiichi Sankyo, and Takeda Pharmaceutical and honoraria from Eisai, Mitsubishi Tanabe Pharma, Takeda Pharmaceutical, Meiji Seika Pharma, Janssen Pharmaceutical, Shionogi, Dainippon Sumitomo Pharma, Daiichi Sankyo, and Pfizer. Yoshimura has received speaker honoraria from Eli Lilly, Janssen, Dainippon Sumitomo, Otsuka, Meiji, Pfizer and Shionogi. Kato has received grant funding from the Japan Society for the Promotion of Science, SENSHIN Medical Research Foundation and the Japan Research Foundation for Clinical Pharmacology and has received speaker honoraria from Dainippon-Sumitomo Pharma, Otsuka, Meiji-Seika Pharma, Eli Lilly, MSD K.K., GlaxoSmithKline, Pfizer, Janssen Pharmaceutical, Shionogi, Mitsubishi Tanabe Pharma, Takeda Pharmaceutical and Ono Pharmaceutical. Azegawa has received speaker honoraria from Eli Lilly, Otsuka Pharmaceutical, and Pfizer. Ueda has received manuscript fees or speaker honoraria from Eli Lilly, Janssen Pharmaceutical, Meiji Seika Pharma, Mitsubishi Tanabe Pharma, MSD, Otsuka Pharmaceutical, Pfizer, Sumitomo Dainippon Pharma, Takeda Pharmaceutical, and Yoshitomi Yakuhin. Edagawa has received speaker honoraria from Eli Lilly, Meiji Seika Pharma, Mitsubishi Tanabe Pharma, MSD, Otsuka Pharmaceutical, Pfizer, Sumitomo Dainippon Pharma, Kyowa and Yoshitomi Yakuhin. Katsumoto has received speaker honoraria from Daiichi Sankyo, Eisai, Eli Lilly, Janssen Pharmaceutical, Kyowa Pharmaceutical, Meiji Seika Pharma, Mitsubishi Tanabe Pharma, MSD, Otsuka Pharmaceutical, Pfizer, Sumitomo Dainippon Pharma, and UCB. Kubota has received consultant fees from Pfizer and Meiji-Seika Pharma and speaker honoraria from Meiji-Seika Pharma, Eli Lilly, Janssen Pharmaceutical, Dainippon Sumitomo Pharma, Mitsubishi Tanabe Pharma, Yoshitomi Yakuhin, Otsuka Pharmaceutical, and Eisai. Goto has received manuscript fees or speaker honoraria from Eli Lilly, Janssen Pharmaceutical, Meiji Seika Pharma, Mitsubishi Tanabe Pharma, MSD, Otsuka Pharmaceutical, and Sumitomo Dainippon Pharma. Hongo has received manuscript fees or speaker honoraria from Eli Lilly, Janssen Pharmaceutical, Kyowa Pharmaceutical, Meiji Seika Pharma, Mitsubishi Tanabe Pharma, Otsuka Pharmaceutical, Pfizer, Shionogi, Sumitomo Dainippon Pharma, and Yoshitomi Yakuhin. Tsuboi has received consultant fees from Pfizer and speaker honoraria from Eli Lilly, Meiji-Seika Pharma, MSD, Janssen Pharmaceutical, Dainippon Sumitomo Pharma, Mitsubishi Tanabe Pharma, Yoshitomi Yakuhin, Mochida Pharmaceutical, Otsuka Pharmaceutical, Kyowa Pharmaceutical, and Takeda Pharmaceutical. Nakagawa has received lecture fees from Pfizer, Eli Lilly, Otsuka, Janssen Pharmaceutical, Mitsubishi Tanabe, Mochida, Dainippon Sumitomo and NTT Docomo and has served on advisory boards for Takeda, Meiji Seika and Tsumura. Kikuchi has received consultant fees from Takeda Pharmaceutical and the Center for Cognitive Behavioral Therapy and Training. Watanabe has received manuscript fees or speaker honoraria from Daiichi Sankyo, Eisai, Eli Lilly, GlaxoSmithKline, Janssen Pharmaceutical, Kyowa Pharmaceutical, Meiji Seika Pharma, Mitsubishi Tanabe Pharma, MSD, Otsuka Pharmaceutical, Pfizer, Shionogi, Sumitomo Dainippon Pharma, Takeda Pharmaceutical, and Yoshitomi Yakuhin; has received research/grant support from Astellas Pharma, Daiichi Sankyo, Eisai, MSD, Mitsubishi Tanabe Pharma, Meiji Seika Pharma, Otsuka Pharmaceutical, Pfizer, Shionogi, and Sumitomo Dainippon Pharma; and is a consultant for Eisai, Eli Lilly, Kyowa Pharmaceutical, Otsuka Pharmaceutical, Pfizer, Sumitomo Dainippon Pharma, Taisho Toyama Pharmaceutical, and Takeda Pharmaceutical.

Additional Editor Comments:

Reviewers' comments:

Reviewer's Responses to Questions

**Comments to the Author**

1. Is the manuscript technically sound, and do the data support the conclusions?

Reviewer #1: Yes

Reviewer #2: Partly

2. Has the statistical analysis been performed appropriately and rigorously? 

Reviewer #1: Yes

Reviewer #2: Yes

3. Have the authors made all data underlying the findings in their manuscript fully available?

Reviewer #1: No

Reviewer #2: Yes

4. Is the manuscript presented in an intelligible fashion and written in standard English?

Reviewer #1: Yes

Reviewer #2: Yes

5. Review Comments to the Author

Reviewer #1: There are a few references missing, which I expect to see in the revised version of the introduction or the discussion:

1) “Clinical and psychopathological features associated with treatment-emergent mania in bipolar-II depressed outpatients exposed to antidepressants” J Affect Disord. 2018 Jul;234:131-138.

2) “Incidence, prevalence and clinical correlates of antidepressant-emergent mania in bipolar depression: a systematic review and meta-analysis” Bipolar Disord. 2018 May;20(3):195-227.

3) “The prevalence and predictors of bipolar and borderline personality disorders comorbidity: Systematic review and meta-analysis” J Affect Disord. 2016 May;195:105-18.

Besides, the authors should also consider the followings:

4) Most of the include covariates dealt with monotherapy, which does not really reflect the real-world setting in BD: please further comment about this issue and consider referencing the suggested reference too: “Prevalence and clinical features associated with bipolar disorder polypharmacy: a systematic review” Neuropsychiatr Dis Treat. 2016 Mar 31;12:719-35.

Reviewer #2: Dear all,

Thank you for submitting this interesting paper about real-world predictors of manic episodes among outpatients with bipolar disorder. Below, I will explain the answers I left, and I will report some minor issues I found:

- The article's title is "Real-world clinical predictors of manic episodes among outpatients with bipolar disorder". However, as described by the authors in their limitations, manic and hypomanic episodes are summed up together. This is a relevant information, and the presented results should be interpreted differently (i.e., 29.1% of the total sample experienced one manic OR one hypomanic episode during the follow-up year), since the two conditions could be quite different in terms of severity and prognosis. The authors should at least: 1) Present this clarification since the beginning; 2) Substitute the concept "manic episode" with "manic/hypomanic episode" throughout the text; 3) Provide, if possible, the exact number of people who experienced a manic episode; 4) Provide an explanation of their choice.

- In the section Matherials and Methods, the authors write that "a questionnaire was administered at 176 of the 1665 outpatient clinics belonging to the JAPC". Could you please further explain why only a portion of clinics was included in the study? If the original project aimed to include a larger portion, I think that this data could be better presented in the "Results".

- Which questionnaire did the psychiatrists complete? Was it a structured/semi-structured one?

- Throughout the text the authors do not refer to any inclusion/exclusion criteria (except for BD NOS). Did the authors apply any other inclusion/exclusion criteria (i.e., age, medical comorbidities, ...)? Please, further explain this point in the section "Matherials and Methods".

- In the section Results, I had some troubles in understanding a couple of ORs. In particular: 1) GAF Score, OR: 1.54 = people with higher scores had 54% higher odds of experiencing a manic/hypomanic episode vs people with lower scores? If so, since you splitted the GAF score variable in four groups, how much low is low? Moreover, if my interpretation is correct, please consider to rephrase the corresponding line in the Results; 2) BD type, OR: 0.55 = I think that is BD-II vs BD-I. Please see the point above.

- Table 1 is not displayed correctly in my copy. Please fix this issue, if possible.

- In the discussion (page 28, line 9), the authors refer to some effect sizes (manic state=4.54; rapid cycling=3.86; mixed features=2.32) as particularly large ones. However, according to the following manuscript (Henian Chen, Patricia Cohen & Sophie Chen (2010) How Big is a Big Odds Ratio? Interpreting the Magnitudes of Odds Ratios in Epidemiological Studies, Communications in Statistics - Simulation and Computation, 39:4, 860-864, DOI: 10.1080/03610911003650383), the considered ORs are smaller in size. Could you please double-check this, also according to the event rate in the exposed/unexposed groups?

- People with personality disorder are more likely to experience a manic/hypomanic episode. Since personality disorders is a noun under which is included a large variety of clinical conditions, is it possible to present a more stratified information (i.e., splitting into cluster A, B, or C), in order to provide a more focused and useful message to the reader?

Thank you,

Best regards

6. PLOS authors have the option to publish the peer review history of their article (what does this mean?). If published, this will include your full peer review and any attached files.

Reviewer #1: **Yes: **Martina Billeci

Reviewer #2: **Yes: **Michele De Prisco

---

## [Author Response · Author response to Decision Letter 0]

29 Oct 2021

Authors’ response to the reviewers:

We are grateful to the reviewers for their critical comments and useful suggestions, which have helped us improve our paper. As indicated in the responses that follow, we have taken all these comments and suggestions into account in the revised version of our paper.

We hope that the revised version of our paper is now suitable for publication in PLOS ONE.

Sincerely,

Norio Yasui-Furukori, MD, PhD

---

## [Editor Report · Decision Letter 1]

17 Dec 2021

Real-world clinical predictors of manic/hypomanic episodes among outpatients with bipolar disorder.

PONE-D-21-18468R1

Dear Dr. Norio,

We’re pleased to inform you that your manuscript has been judged scientifically suitable for publication and will be formally accepted for publication once it meets all outstanding technical requirements.

Kind regards,

Michele Fornaro

Academic Editor

PLOS ONE
---

## [Editor Report · Acceptance letter]

22 Dec 2021

PONE-D-21-18468R1 

Real-world clinical predictors of manic/hypomanic episodes among outpatients with bipolar disorder. 

Dear Dr. Norio:

I'm pleased to inform you that your manuscript has been deemed suitable for publication in PLOS ONE. Congratulations! Your manuscript is now with our production department. 

Kind regards, 

on behalf of

Dr. Michele Fornaro 

Academic Editor

PLOS ONE